# Ultra-massive fluid transfusion in adult liver transplant recipients: A single center observational study

Hugh Slifirski[1](ID), Nattaya Raykateeraroj(ID)[2](ID), Angelica Armellini[1], Riley Hazard[1], Jordan Zalcman[1], Junyan Zhao[1], Zac Tran[1], Peter Le(ID)[1], Wendell Zhang[1], Michael Fink(ID)[3,4], Marcos Vinicius Perini[3,4], Anoop N. Koshy[5], Dong-Kyu Lee(ID)[6], Laurence Weinberg(ID)[1,7]*

1 Department of Anaesthesia, Austin Health, Heidelberg, Victoria, Australia, 2 Department of Anesthesiology, Faculty of Medicine Siriraj Hospital, Mahidol University, Bangkok, Thailand, 3 Department of Surgery (Austin Precinct), The University of Melbourne, Melbourne, Victoria, Australia, 4 Victorian Liver Transplant Unit, Austin Health, Melbourne, Victoria, Australia, 5 Department of Cardiology, Austin Health, Heidelberg, Victoria, Australia, 6 Department of Anaesthesiology and Pain Medicine, Dongguk University Ilsan Hospital, Goyang, Korea (the Republic of), 7 Department of Critical Care, The University of Melbourne, Melbourne, Victoria, Australia

☯ These authors contributed equally to this work (co-first authors) (HS and NR).
* laurence.weinberg@austin.org.au

## Abstract

### Introduction

Patients undergoing liver transplantation may require large volumes of fluid to maintain hemodynamic stability and treat coagulopathy. This study aimed to determine the prevalence of ultra-massive fluid transfusion and to examine its association with clinical outcomes. We defined an ultra-massive fluid transfusion *a priori* as a transfusion volume of >20 liters of crystalloids, colloids, blood and blood products administered intraoperatively and within the first 24 hours postoperatively.

### Methods

This single-center retrospective observational study included all adult patients who underwent an orthotopic liver transplant and received an ultra-massive fluid transfusion. The primary aim was to determine the prevalence of ultra-massive fluid transfusion in patients undergoing liver transplantation. Secondary objectives included evaluating the effect of the total volume of fluid and packed red blood cell transfusions on postoperative complications, mechanical ventilation hours, intensive care unit and hospital length of stay, and mortality.

### Results

Of the 844 liver transplantation procedures, 81 (9.6%) required an ultra-massive fluid transfusion with a median transfusion volume of 36.8 liters (IQR: 31.2–48.7). Each additional liter of fluid administered during surgery was associated with an additional

**Data availability statement:** All relevant data are within the paper and its Supporting information files.

**Funding:** The author(s) received no specific funding for this work.

**Competing interests:** The authors have declared that no competing interests exist.

stay of 0.47 days in intensive care (95%CI: 0.18–0.76, $p = 0.003$). Each additional unit of packed red blood cells administered during surgery was associated with an additional 12.8 hours of mechanical ventilation (95%CI: 3.12–22.43, $p = 0.014$) and 1.0 additional day in intensive care (95%CI: 0.27–1.79, $p = 0.012$). Neither ultra-massive fluid transfusion nor packed red blood cell transfusions were associated with increased complications.

### Conclusion

Approximately one in ten liver transplantation patients required an ultra-massive fluid transfusion. While ultra-massive fluid transfusion was associated with prolonged recovery, it was not associated with an increased risk of complications or mortality.

---

### Introduction

Ultra-massive transfusion (UMT), defined as the transfusion of ≥20 units of packed red blood cell (PRBC) products within 24 hours [1], is frequently used in trauma patients with hemorrhagic shock, which is associated with poor postoperative outcomes and increased mortality [2–5]. However, UMT is also employed in other major procedures, including solid organ transplantation, where 84% of the UMT cases among solid organ transplant patients involve liver transplantation (LT) [6]. Large-volume resuscitation during liver transplantation (LT) is necessitated by the vascularity of the liver [7], the hyperdynamic state of end-stage liver disease [8], portal hypertension [9], and the complex hemostatic abnormalities that increase the risk of bleeding and clotting [8,10–12]. These challenges, compounded by dilutional coagulopathy, make LT a complex procedure that requires substantial transfusion support [9,13].

The transfusion of blood and blood products during LT is linked to increased morbidity and mortality [14–17]. To minimize allogeneic transfusion, various strategies used in other major surgeries such as preoperative anemia treatment, restrictive transfusion triggers, point-of-care coagulation monitoring, antifibrinolytics, factor concentrates, and cell salvage have been implemented in LT [8,9,17–19]. Despite these efforts, massive transfusions remain a concern for LT. Therefore, we investigated the prevalence of ultra-massive fluid transfusion (UMFT) and its association with postoperative complications, mortality, ventilation time, and length of intensive care unit (ICU) and hospital stay. Given that there is no established definition of UMFT in the context of liver transplantation, we defined an UMFT *a priori* as a combined transfusion volume of >20 liters of crystalloids, colloids, blood and blood products administered intraoperatively and within 24 hours postoperatively.

### Materials and methods

#### Ethical approval

This case series was retrospectively registered in the Australian-New Zealand Clinical Trials Registry (no: 12624000499583) on April 23, 2024. Ethical approval was

obtained from the Austin Health Human Research Ethics Committee (HREC/105884/Austin-2024) on March 25, 2024. Data collection for the first participant commenced on March 26, 2024. The last patient was enrolled on April 17, 2024. Data collection for all participants was completed on April 19, 2024. The ethics committee granted a waiver of informed consent owing to the retrospective nature of the study and the use of deidentified data. No changes were made to the original study protocol at any stage, and data analysis commenced only after ethical approval was obtained.

## Study population, data sources, and variables

After human research ethics committee approval, we conducted this single-center retrospective observational study that included all adult patients (≥ 18 years) who underwent orthotopic LT at Austin Health between November 14, 2009, and April 1, 2023. Austin Health, Melbourne, Australia, is a university teaching hospital that is home to the Victorian Liver Transplant Unit. The unit has performed over 1,800 LTs to date and currently conducts approximately 100 LTs annually using only deceased donor grafts for adult recipients. Patients were eligible for inclusion if they received a combined transfusion volume of >20 liters of crystalloids, colloids, blood and blood products intraoperatively and within 24 hours postoperatively.

Coagulation management was directed by point-of-care thromboelastography (TEG), which was performed at the following standardized time points: induction of anesthesia, hourly during the hepatic dissection phase of surgery, every 30 minutes during the anhepatic phase, immediately before and five minutes after reperfusion, then every 30 minutes for the subsequent hour, and thereafter hourly during the neohepatic phase until surgical closure. In the intensive care unit (ICU), TEG was conducted every six hours.

All patients received invasive hemodynamic monitoring utilizing a pulmonary artery catheter, a central venous catheter, as well as radial and femoral arterial lines. Transesophageal echocardiography (TEE) was employed in all patients, unless contraindicated, to provide real-time, dynamic assessment of cardiac function, volume status, and intraoperative complications. TEE is used in our centre to facilitate the prompt identification of the underlying causes of hemodynamic instability, including hypovolemia, right ventricular dysfunction, or embolic events.

Data were collected from electronic medical records of patients and the prospectively maintained database of the transplant unit. The extracted preoperative variables included demographic data, comorbidities, indications for LT, model for end-stage liver disease (MELD) score, and baseline laboratory data measured immediately prior to transplantation. Perioperative data included operation type, surgery duration, use of veno-venous bypass, intraoperative vasoactive medications, and fluid and blood administration. We collected data on the volumes of crystalloids, colloids, cell-saved blood, PRBCs, fresh frozen plasma (FFP), cryoprecipitate, and platelets administered intraoperatively and up to 24 hours postoperatively. The number of units of all blood products was also recorded. Additionally, intraoperative blood gas analyses, routine biochemistry, lactate concentrations, and coagulation tests were performed at each stage of the procedure.

Postoperative outcomes included all perioperative complications, which were systematically collected and graded using the Clavien-Dindo (CD) classification system [20]. Liver transplant-specific complications included primary graft non-function, long-term graft failure, bile leakage, hepatic artery or vein thrombosis, and liver abscess. We also collected data on mechanical ventilation hours, ICU and hospital length of stay (LOS), and mortality, including on-table, in-hospital, 30-day, 1-year, 5-year, and overall mortality.

## Key outcomes

The primary aim of this study was to determine the prevalence of UMFT in patients undergoing LT. The secondary objectives were to evaluate the effect of the total volume of fluid and the number of PRBC units transfused during the intraoperative period and the first 24 hours postoperatively on postoperative complications, mechanical ventilation hours, ICU and hospital LOS, and mortality. Additionally, this study evaluated the PRBC: FFP: platelet transfusion ratios used in the intraoperative and perioperative periods and their association with postoperative complications and mortality.

## Statistical analysis

All statistical analyses were performed using R 4.3.2 (R Core Team, 2023, Vienna, Austria). Continuous variables were assessed for normality using Shapiro-Wilk test and visual inspection of the Q-Q plots. Descriptive statistics are presented as mean±standard deviation (SD), or median (1st – 3rd quartiles). Categorical variables are reported as the number of cases (percentage), with estimated values presented alongside 95% confidence intervals (CI). Statistical $p$ value was set at $p < 0.05$, based on null hypothesis significance testing, and effect sizes were reported to quantify the magnitude of associations. Extreme values were checked using the 1st and 3rd quartiles and a two-fold interquartile range (IQR) step. All extreme values were compared with the original data source, and their clinical relevance was discussed with two authors (LW, NR). Ultimately, none of the extreme values was modified.

Variables with missing rates higher than 5% were identified and missing patterns were evaluated (S1 Table). Most missing values were associated with specific LT surgical stages and were closely related to other variables such as baseline laboratory values, indicating that their absence did not critically affect the outcomes. Missing values were excluded from the analysis. All statistical analyses were performed on a case-by-case basis, and exclusions were made where appropriate.

Unadjusted relationships between the outcome variables, including ventilation time, ICU and hospital LOS, and the volume of UMFT and the number of units of PRBCs transfused were assessed using Pearson's or Spearman's correlation analysis. The relationships between the outcomes and the volume of UMFT were analyzed separately for intraoperative, postoperative (first 24 hours), and combined intraoperative and postoperative periods. These analyses, along with the units of PRBCs transfusion and the ratio of FFP, platelet and cryoprecipitate transfusions, were further evaluated using linear or logistic regression models, depending on the nature of the response variables. During regression analysis, adjustments were made for potential confounders, including age, body mass index (BMI), MELD score, preoperative prothrombin time (PT), albumin, platelet count, hemoglobin, and fibrinogen values, the use of veno-veno bypass, piggyback technique, surgery duration, and the highest dose of intraoperative noradrenaline that was administered. Baseline biochemical parameters such as bicarbonate, calcium, lactate, and D-dimer were also included to ensure a comprehensive evaluation of the associations between transfusion practices and key postoperative outcomes, including complications, ventilation hours, ICU and hospital LOS, and mortality.

The associations between intraoperative and postoperative fluid/transfusion volumes and postoperative recovery outcomes were explored with two regression models, adjusted for perioperative covariates. Models for intraoperative and postoperative variables of fluid or transfusion were fitted simultaneously to assess their independent associations with clinical outcomes. A sensitivity analysis, that we termed the "trajectory model", quantified the net difference in fluid volumes between the intraoperative and postoperative 24-hour periods. The net change in postoperative fluid or transfusion volume administered, that we termed the "resuscitation shift", was compared to the volume of fluid administered intraoperatively.

## Results

The study initially enrolled 844 patients who underwent LT between November 14, 2009, and April 1, 2023. Among these, 81 patients who received an UMFT were included in our analysis (Fig 1).

### Clinical characteristics

Among this cohort (Table 1), 60.5% of the patients were male, with a mean age of 54.2±10.9 years and a median MELD score of 20 (IQR: 14–26). The most common indications for LT were alcoholic cirrhosis (17.5%), primary sclerosing cholangitis (15.0%), and non-alcoholic steatohepatitis (NASH) (13.8%). Single liver transplant was performed in 96.3% of cases and 3.7% of the cohort involved multi-organ transplantation (both liver and kidney). Redo liver transplantation accounted for 9.9% of cases, and veno-venous bypass was used in 4.9% of the procedures.

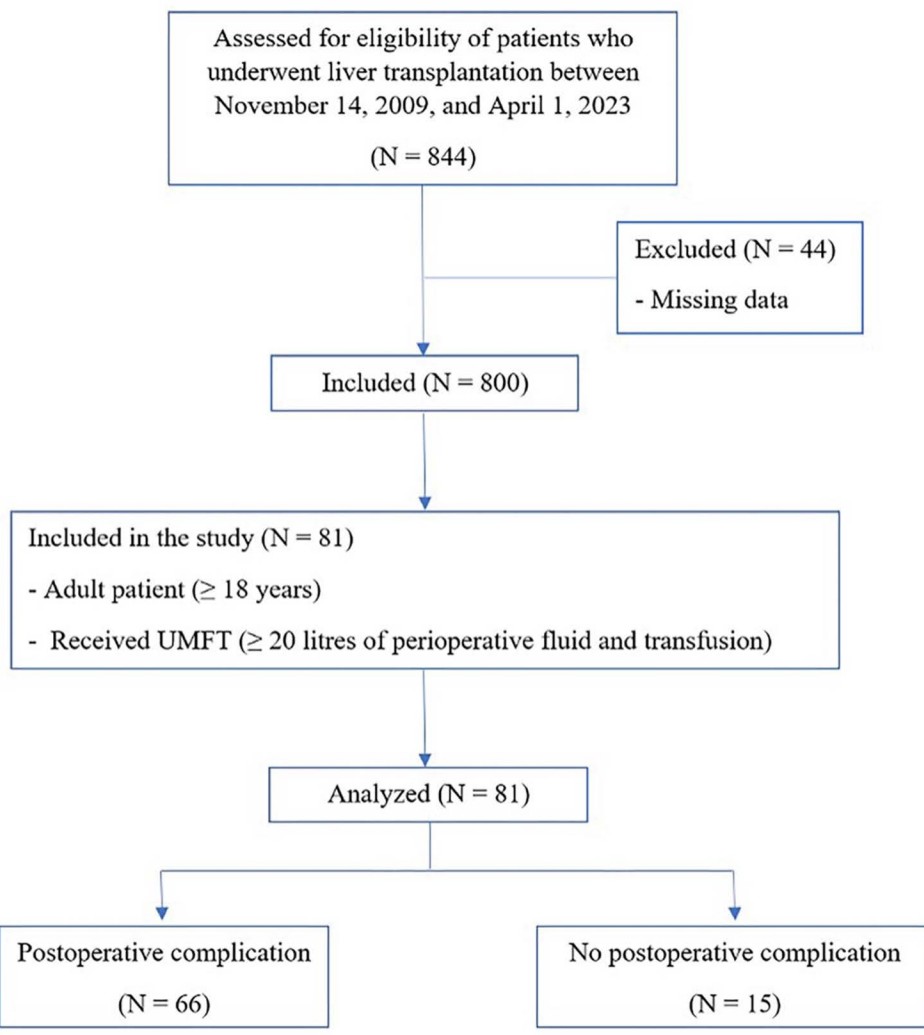

**Fig 1. Study flow diagram.**

## Intraoperative management

The median numbers of intraoperative transfusions were 13 units of PRBCs (IQR: 10–17 units), 6 units of FFP (IQR: 4–8), 15 units of cryoprecipitate (IQR: 10–21), and 3 units of platelets (IQR: 2–4). Fluid management, blood transfusion and use of vasoactive medications during surgery and the first 24 hours postoperatively are summarized in Tables 2 and 3. Detailed preoperative, baseline, and intraoperative laboratory results are presented in S2 Table. The timing of fluid, blood and blood product administration in presented graphically in Fig 2. All 81 patients received tranexamic acid (2 grams) during the dissection or anhepatic phases. Five patients (6.2%) had evidence of hyperfibrinolysis on point-of-care TEG during the neohepatic phase, and each received an additional 2 grams of tranexamic acid.

## Postoperative complications

As summarized in Table 4, 66 patients (81.5%) developed a postoperative complication, including 64.2% who experienced more than one, while 15 (18.5%) had none. Among those with complications, the highest grade of complications, based

**Table 1. Demographic and clinical characteristics of the study population.**

| Variable (n = 81) | Values |
|---|---|
| **Demographic** | |
| Age (years) | 54.2 ± 10.9 |
| Male | 49 (60.5%) |
| BMI (kg/m²) | 30.0 ± 9.1 |
| MELD score | 20 (14–26) |
| Diabetes | 3 (3.8%) |
| Hypertension | 9 (11.2%) |
| Preoperative antiplatelet medication (aspirin) | 3 (3.8%) |
| Other preoperative anticoagulation therapy | 0 (0%) |
| **Etiology** | |
| Alcoholic liver cirrhosis | 14 (17.5%) |
| Primary sclerosing cholangitis | 12 (15%) |
| Non-alcoholic steatohepatitis | 11 (13.8%) |
| Hepatitis C virus cirrhosis | 9 (11.2%) |
| Hepatitis B virus cirrhosis | 7 (8.8%) |
| Hepatocellular carcinoma | 7 (8.8%) |
| Autoimmune liver cirrhosis | 5 (6.2%) |
| Polycystic liver disease | 3 (3.8%) |
| Primary biliary cirrhosis | 2 (2.5%) |
| Hemochromatosis | 1 (1.3%) |
| Other etiology[1] | 9 (11.2%) |
| **Type of transplant** | |
| Single liver transplant | 78 (96.3%) |
| Redo liver transplantation | 8 (9.9%) |
| Multi-organ transplant (liver with kidney) | 3 (3.7%) |
| **Surgical technique** | |
| Caval-sparing (piggyback) | 71 (87.65%) |
| Veno-venous bypass use | 4 (4.9%) |
| Side-to-side cavo-cavostomy | 3 (3.7%) |
| Caval replacement | 3 (3.7%) |
| **Duration of surgery (minutes)** | 699.1 ± 137.4 |

Data are presented as number (%), mean ± SD or median (interquartile range).

Abbreviation: BMI, body mass index; MELD, model for end-stage liver disease.

[1]Other etiologies include alpha 1-antitrypsin deficiency, cryptogenic, hepatitis A virus infection, hepatitis D virus infection, hepatic artery thrombosis, recurrent pyogenic cholangitis, redo for graft cirrhosis, and sarcoidosis.

on the CD classification, was Grade II (39.5%), followed by Grade IIIb (38.3%), Grade IVb (1.2%), and Grade V (2.5%). Surgery-specific complications occurred in 39 patients (48.1%), with the most frequent being bleeding (14.8%), hepatic artery or vein thrombosis (8.6%), and bile leaks (6.2%).

Thirty-three patients (40.7%) required re-operations, including for generalized bleeding (12.3%) and other complications such as bile duct issues, perforations, and vascular problems (19.7%). Seven patients (8.6%) required re-transplantation due to primary early graft non-function, with no graft failures occurring beyond 30 days post-transplant. The other post-operative complications are listed in S3 Table. Intraoperative TEE was performed in 65 patients (80%). Intraoperative

**Table 2. Fluid management and transfusion during surgery and the first 24 hours postoperatively (n = 81).**

| Variable | Measurement |
|---|---|
| **Total fluid volume (L)** (combined transfusion volume of crystalloids, colloids, blood, and blood products administered intraoperatively and within 24 hours postoperatively) | 31.9 ± 16.3 |
| **Intraoperative fluid volumes** | |
| Plasma-Lyte (L) (n = 81) | 16.4 ± 14.7 |
| Albumin 20% (L) (n = 81) | 2.5 (2.0 – 3.2) |
| Organ donor blood (L) (n = 28) | 0.9 (0.7 – 0.9) |
| Cell saver (L) (n = 77) | 8.4 ± 5.2 |
| Total blood bank product (L) | 8.1 ± 3.8 |
| **Total number of units of blood products** | |
| PRBCs (units) (n = 81) | 13 (10 – 17) |
| FFP (units) (n = 81) | 6 (4 – 8) |
| Cryoprecipitate (units) (n = 81) | 15 (10 – 21) |
| Platelets (units) (n = 81) | 3 (2 – 4) |
| **Postoperative transfusion (first 24 hours)** | |
| Total fluid (L) | 2.0 ± 1.1 |
| Plasma-Lyte (L) (n = 76) | 0.5 ± 0.6 |
| Albumin 20% (L) (n = 37) | 0.2 (0.1 – 0.3) |
| Albumin 4% (L) (n = 50) | 1.0 (0.5 – 1.5) |
| Dextrose (L) (n = 52) | 1.0 ± 0.22 |
| Total blood bank product (L)[1] | 1.7 ± 2.2 |
| **Total number of units of blood products** | |
| PRBCs (units) (n = 58) | 2 (1 – 5) |
| FFP (units) (n = 33) | 2 (2 – 5) |
| Cryoprecipitate (units) (n = 41) | 10 (10 – 20) |
| Platelet (units) (n = 42) | 1 (1 – 2) |

Data are expressed as mean ± SD or median (IQR).

(n) indicates the number of patients who received the specified treatment or intervention.

[1]Total blood bank products include all transfused blood products (e.g., PRBCs, FFP, cryoprecipitate, platelets) but exclude cell saver blood and organ donor blood.

Abbreviation: PRBCs, packed red blood cells; FFP, fresh frozen plasma.

**Table 3. Vasopressor use during surgery and the first 24 hours postoperatively (n = 81).**

| Variable (n = 81) | Measurement |
|---|---|
| **Intraoperative management** | |
| Norepinephrine administration | 81 (100%) |
| Total norepinephrine use (mg) | 7.5 ± 4.3 |
| Highest norepinephrine dose (μg/min) | 28.0 ± 13.1 |
| Vasopressin administration | 21 (25.9%) |
| Epinephrine administration | 14 (17.3%) |
| Methylene blue administration | 14 (17.3%) |
| **Postoperative management (first 24 hours)** | |
| Highest norepinephrine dose (μg/min) (n = 77) | 8 (3 – 14) |
| Highest epinephrine dose (μg/min) (n = 5) | 2.4 (2.0 – 2.6) |
| Vasopressin administration | 27 (33.3%) |

Data are expressed as number (%), mean ± SD or median (interquartile range).

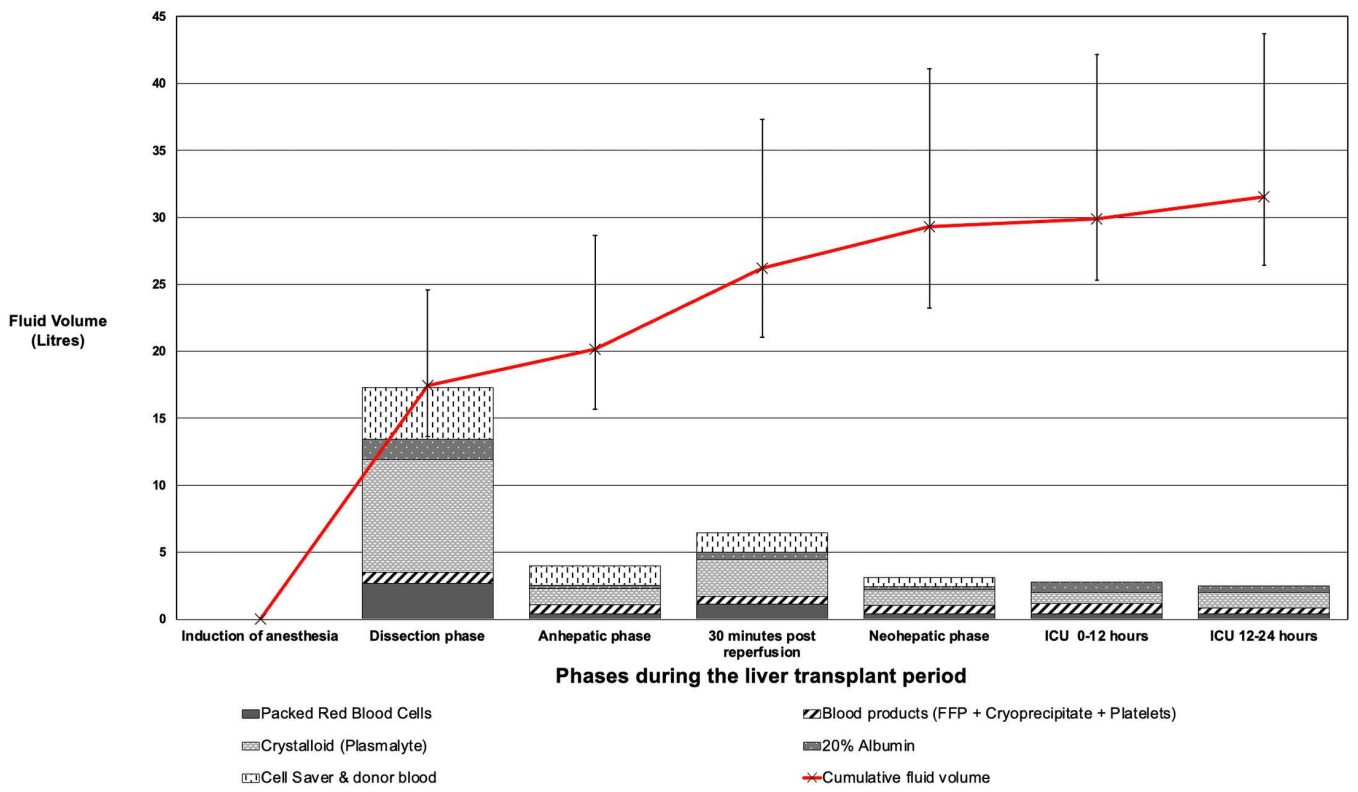

**Fig 2. Fluids, blood and blood product administration during surgery and for the first 24 hours postoperatively.**

evidence of right ventricular (RV) dysfunction was identified in six patients; of these, one patient died intraoperatively, while RV function normalized postoperatively in the remaining five patients.

### Associations between fluid/transfusion volumes and PRBCs transfusions with postoperative outcomes

The calculated unadjusted correlations revealed significant associations between key postoperative outcomes—including mechanical ventilation hours, ICU LOS, and hospital LOS—and total fluid/transfusion volume, as well as transfused PRBC units. Similarly, unadjusted logistic regression showed significant associations between total fluid/transfusion volume and PRBC units with postoperative complications. Additionally, higher PRBC transfusions were linked to lower odds of reoperation, multiple complications, and severe complications, while no significant associations were observed for surgery-specific complications or 30-day mortality (Table 5).

The intraoperative and postoperative administration of fluids, PRBC units, and other blood products demonstrated statistically significant associations with adverse postoperative recovery metrics, including duration of mechanical ventilation and length of ICU and hospital stays. These are presented in the S4 Table. The magnitude of association was more pronounced for postoperative fluid volumes, PRBC units, and transfusions compared to their intraoperative counterparts, as evidenced by larger regression coefficients. The negative coefficients observed for resuscitation shift parameters suggested that increased postoperative fluid administration and transfusion (represented by smaller shift values) correlated with poorer clinical outcomes, thus highlighting the potential deleterious effects of excessive fluid or blood product administration during the early postoperative period. The analysis of the discrete fluid volumes and transfusion quantities administered during intraoperative, or postoperative 24-hour periods is presented in S5 Table.

**Table 4. Postoperative complications and reasons for reoperation.**

| Complications | All patients (n=81) |
|---|---|
| **Number of patients with any postoperative complications** | 66 (81.5%) |
| **Number of complications per patient** | |
| 1 complication | 14 (17.3%) |
| 2 complications | 17 (21.0%) |
| 3 complications | 12 (14.8%) |
| 4 complications | 9 (11.1%) |
| 5 complications | 14 (17.3%) |
| **Highest Clavien-Dindo grade per patient** | |
| Clavien-Dindo Grade II | 32 (39.5%) |
| Clavien-Dindo Grade IIIb | 31 (38.3%) |
| Clavien-Dindo Grade IVb | 1 (1.2%) |
| Clavien-Dindo Grade V | 2 (2.5%) |
| **Surgery-specific complications (n=39)** | |
| **Bleeding requiring blood products** | |
| Clavien-Dindo Grade II | 2 (2.5%) |
| Clavien-Dindo Grade IIIb | 10 (12.3%) |
| **Bile leak** | |
| Clavien-Dindo Grade II | 1 (1.2%) |
| Clavien-Dindo Grade IIIb | 4 (4.9%) |
| **Hepatic artery/vein thrombosis** | |
| Clavien-Dindo Grade II | 3 (3.7%) |
| Clavien-Dindo Grade IIIb | 4 (4.9%) |
| **Liver abscess** | |
| Clavien-Dindo Grade II | 1 (1.2%) |
| **Other complications[1]** | |
| Clavien-Dindo Grade II | 5 (6.2%) |
| Clavien-Dindo Grade IIIa | 1 (1.2%) |
| Clavien-Dindo Grade IIIb | 8 (9.9%) |
| **Reoperations or interventions (n=33)** | |
| **Bleeding** | |
| Clavien-Dindo Grade IIIb | 10 (12.3%) |
| **Other operations[2]** | |
| Clavien-Dindo Grade IIIa | 1 (1.2%) |
| Clavien-Dindo Grade IIIb | 15 (18.5%) |
| **Primary graft non-function/early allograft dysfunction** | |
| Clavien-Dindo Grade IIIb | 7 (8.6%) |
| Graft loss beyond 30 days post-transplant | 0 |

Data are presented as number and percentage of patients with each postoperative complication.

[1]Other complications include bowel perforation; gastric perforation; anastomotic biliary stricture; bile leak; biloma; cholangitis; esophageal perforation; perihepatic collection; portal vein stenosis; hepatic collections; sub-hepatic collection; hepatic artery dissection.

[2]Other operations include bile duct reconstruction; bile duct stenosis requiring stenting; biliary anastomosis; small bowel resection for incarcerated hernia; hemicolectomy; end ileostomy; Roux-en-Y; choledocojejunostomy; hepatic artery thrombectomy; perihepatic collection/ascites; repair of biliary anastomosis; repair of the portal vein and delayed biliary anastomosis; and wound dehiscence.

**Table 5. Unadjusted linear regression and logistic regression analysis of the association between PRBCs transfused and total fluid/transfusion volumes with postoperative outcomes.**

| Linear regression analysis | PRBCs transfusion (unit) | | Total fluid and blood products (mL) | |
| --- | --- | --- | --- | --- |
| | Correlation coefficient (r) | *p*-value | Correlation coefficient (r) | *p*-value |
| **Recovery outcomes** | | | | |
| Mechanical ventilation (hours) | 0.362 | 0.001* | 0.271 | 0.017* |
| ICU LOS (days) | 0.411 | < 0.001* | 0.368 | 0.001* |
| Hospital LOS (days) | 0.409 | < 0.001* | 0.379 | 0.001* |
| **Logistic regression analysis** | PRBCs transfusion (unit) | | Total fluid and blood products (mL) | |
| | Odds ratio (OR) | *p*-value | Odds ratio (OR) | *p*-value |
| **Complications** | | | | |
| Surgery-specific complications | 0.149 | 0.184 | 0.053 | 0.638 |
| Reoperation or interventions | 0.465 | < 0.001* | 0.332 | 0.002* |
| Number of complications | 0.326 | 0.003* | 0.208 | 0.063 |
| Highest Clavien-Dindo grade | 0.285 | 0.010* | 0.104 | 0.354 |
| 30-day mortality | 0.033 | 0.771 | 0.056 | 0.619 |

Data are presented as correlation coefficients (r) from linear regression for recovery outcomes and odds ratios (ORs) from logistic regression for complications, with corresponding *p*-values in parentheses.

*Indicates statistical significance (*p* < 0.05).

Abbreviation: PRBCs, packed red blood cells; ICU, intensive care unit; LOS, length of stay.

There was a statistically significant association with PRBC unit counts and surgery-specific complications and severe postoperative complications (CD grade ≥ 3). As shown in Table 6, postoperative PRBCs transfusions had the most substantial effect on recovery outcomes, significantly increasing ventilation duration (47.3 hours/unit, *p* < 0.001), ICU LOS (3.3 days/unit, *p* < 0.001), and hospital LOS (4.3 days/unit, *p* = 0.005), with a similar pattern observed for total fluid/transfusion volume. Postoperative PRBC transfusion was also associated with a lower risk of surgery-specific complications (odds ratio [OR] = 0.35, 95% confidence interval [CI]: 0.13–0.97, *p* = 0.043), potentially reflecting therapeutic benefits from the

**Table 6. Adjusted linear regression coefficients for associations between PRBCs transfusion and total fluid/transfusion volumes with recovery outcomes.**

| Recovery outcomes | Intraoperative | | Postoperative | | Overall | |
| --- | --- | --- | --- | --- | --- | --- |
| | Coefficient (95% CI) | *p*-value | Coefficient (95% CI) | *p*-value | Coefficient (95% CI) | *p*-value |
| **PRBCs transfusion (units)** | | | | | | |
| Mechanical ventilation (hours) | 12.77 (3.12 – 22.43) | 0.014* | 47.34 (31.01 – 63.67) | < 0.001* | 16.7 (9.63 – 23.76) | < 0.001* |
| ICU LOS (days) | 1.03 (0.27 – 1.79) | 0.012* | 3.33 (1.93 – 4.74) | < 0.001* | 1.27 (0.7 – 1.84) | < 0.001* |
| Hospital LOS (days) | 2.01 (0.8 – 3.22) | 0.002* | 4.26 (1.48 – 7.04) | 0.005* | 2.25 (1.29 – 3.21) | < 0.001* |
| | Coefficient (95% CI) | *p*-value | Coefficient (95% CI) | *p*-value | Coefficient (95% CI) | *p*-value |
| **Total fluid and blood products (mL)** | | | | | | |
| Mechanical ventilation (hours) | 6.01 (2.38 – 9.63) | 0.003* | 52.84 (30.51 – 75.18) | < 0.001* | 6.4 (3.11 – 9.7) | 0.001* |
| ICU LOS (days) | 0.47 (0.18 – 0.76) | 0.003* | 3.95 (2.13 – 5.77) | < 0.001* | 0.5 (0.24 – 0.76) | 0.001* |
| Hospital LOS (days) | 0.82 (0.36 – 1.27) | 0.001* | 4.31 (0.67 – 7.95) | 0.026* | 0.84 (0.41 – 1.27) | < 0.001* |

Data are presented as correlation coefficients (r) with 95% confidence intervals (CIs) from linear regression for recovery outcomes, along with corresponding *p*-values.

*Indicates statistical significance (*p* < 0.05).

Abbreviation: PRBCs, packed red blood cells; ICU, intensive care unit; LOS, length of stay.

timely correction of postoperative anemia or active hemorrhage. However, only 15.8% of surgery-specific complications in this cohort were attributable to bleeding, necessitating more granular data to clarify the role of postoperative transfusion in mitigating such outcomes.

Table 7 examines the associations between PRBC transfusion and total fluid/transfusion volumes with postoperative complications. Neither total fluid and blood product volumes nor PRBC transfusions were significantly associated with surgery-specific complications. PRBC transfusions showed a borderline trend for severe complications (CD grade ≥ 3) overall (OR 1.32, $p = 0.056$); however, this did not reach statistical significance. The analysis of the other outcomes is detailed in S6 and S7 Tables.

### Relationship between the ratios of blood and other blood products, and postoperative outcomes

Table 8 explores the relationships between the interaction ratios (FFP: PRBC, platelet: PRBC and cryoprecipitate: PRBCs) and postoperative outcomes. Both FFP: PRBC and platelet: PRBC ratios were significantly associated with prolonged mechanical ventilation hours, with platelet: PRBC ratios showing the strongest effect (57.12 hours, $p = 0.001$), compared to FFP: PRBC ratios (29.84 hours, $p = 0.005$). Similarly, both ratios were significantly associated with increased ICU LOS,

**Table 7. Adjusted logistic regression for associations between PRBC transfusion and total fluid/transfusion volumes with postoperative complications.**

| Complications | Intraoperative | | Postoperative | | Overall | |
|---|---|---|---|---|---|---|
| | OR (95% CI) | p-value | OR (95% CI) | p-value | OR (95% CI) | p-value |
| **PRBCs transfusion (units)** | | | | | | |
| Presence of surgery-specific complication | 1 (0.88 – 1.13) | 0.99 | 0.48 (0.22 – 1.03) | 0.059 | 0.97 (0.87 – 1.08) | 0.542 |
| Number of complications ≥ 3 | 1.06 (0.93 – 1.23) | 0.381 | 0.91 (0.57 – 1.44) | 0.686 | 1.05 (0.93 – 1.18) | 0.460 |
| Severe complications (Clavien-Dindo grade ≥ 3) | 1.2 (0.97 – 1.48) | 0.087 | 1.36 (0.97 – 1.91) | 0.077 | 1.32 (0.99 – 1.75) | 0.056 |
| Reoperation: bleeding | 0.76 (0.54 – 1.08) | 0.122 | 1.25 (0.84 – 1.85) | 0.267 | 0.87 (0.66 – 1.13) | 0.291 |
| **Total fluid and blood products (mL)** | | | | | | |
| Development of a surgery-specific complication | 0.98 (0.93 – 1.03) | 0.357 | 0.58 (0.3 – 1.14) | 0.112 | 0.97 (0.92 – 1.02) | 0.293 |
| Number of complications ≥ 3 | 1.01 (0.96 – 1.06) | 0.799 | 0.95 (0.56 – 1.6) | 0.841 | 1.01 (0.96 – 1.06) | 0.798 |
| Severe complications (Clavien-Dindo grade ≥ 3) | 1.01 (0.96 – 1.07) | 0.594 | 1.41 (0.9 – 2.19) | 0.13 | 1.02 (0.97 – 1.07) | 0.460 |
| Reoperation due to bleeding | 0.71 (0.34 – 1.5) | 0.368 | 1.23 (0.76 – 1.97) | 0.395 | 0.83 (0.65 – 1.06) | 0.130 |

Data are presented as odds ratios (ORs) with 95% confidence intervals (CIs) from logistic regression for complication outcomes, along with corresponding $p$-values.

*Indicates statistical significance ($p < 0.05$).

Abbreviation: PRBCs, packed red blood cells.

**Table 8. Adjusted relationship between the ratio of FFP, platelet and cryoprecipitate to PRBC and the development of postoperative complications.**

| Outcomes | FFP: PRBC ratio | | Platelet: PRBC ratio | | Cryoprecipitate: PRBC ratio | |
|---|---|---|---|---|---|---|
| | Coefficient (95%CI) | p-value | Coefficient (95%CI) | p-value | Coefficient (95%CI) | p-value |
| Mechanical ventilation (hours) | 29.84 (10.25 – 49.44) | 0.005* | 57.12 (27.66 – 86.58) | 0.001* | 10.89 (5.10 – 16.69) | 0.001* |
| ICU LOS (days) | 1.99 (0.34 – 3.65) | 0.024* | 3.18 (0.72 – 5.64) | 0.016* | 0.53 (0.01 – 1.06) | 0.055 |
| Hospital LOS (days) | 4.01 (1.29 – 6.73) | 0.007* | 1.65 (−3.04 – 6.34) | 0.495 | 0.89 (0.04 – 1.73) | 0.047* |

Data are presented as correlation coefficients (r) with 95% confidence intervals (CIs) from linear regression for recovery outcomes, along with corresponding $p$-values.

*Indicates statistical significance ($p < 0.05$).

Abbreviation: FFP, fresh frozen plasma; PRBCs: packed red blood cells; ICU: intensive care unit; LOS: length of stay.

with platelet: PRBC ratios (3.18 days, *p* = 0.016) showing a larger effect than FFP: PRBC ratios (1.99 days, *p* = 0.024). However, only FFP: PRBC ratios were significantly associated with an increase in hospital LOS (4.01 days, *p* = 0.007), whereas platelet: PRBC ratios showed no significant association (*p* = 0.495).

A higher cryoprecipitate: PRBC ratio was associated with a significantly reduction in duration of mechanical ventilation (coefficient = –99.42, 95% CI: –189.85 to –8.99, *p* = 0.039). There was a significant interaction between cryoprecipitate: PRBC ratio (coefficient = 10.89, 95% CI: 5.10 to 16.69, *p* = 0.001), indicating that the beneficial effect of a higher cryoprecipitate ratio on reducing ventilation time diminished as PRBC transfusion volume increased. The cryoprecipitate ratio alone was not significantly associated with length of hospital stay (*p* = 0.576). The complete analyses, including details of individual effects of PRBCs, FFP, cryoprecipitate and platelets, are provided in S8–S11 Tables.

## Discussion

### Key findings

The study confirmed that 1 in 10 patients undergoing LT required an UMFT. Our analysis showed that higher fluid and transfusion volumes, including a high FFP: PRBC and platelet: PRBC ratio, were associated with prolonged recovery, as reflected in extended mechanical ventilation times, longer ICU stays, and increased hospital LOS, but not an increased risk of complications or mortality. We observed a consistent pattern of a PRBC: FFP: platelet transfusion ratio of approximately 4: 2: 1 in contrast to a 1: 1: 1 ratio commonly used in trauma settings. This ratio was effective in managing blood loss, correcting coagulopathy, and supporting stable recovery in LT patients.

### Relationship to the existing literature

Few studies have examined the association of UMFT and outcomes in the setting of LT. Liver transplantation presents distinct challenges due to the unique pathophysiology of chronic liver disease [7,8,10–12] and the complexity of the surgical technique [21,22]. Allogeneic transfusions during LT are associated with increased morbidity and mortality [14–16], especially when the transfusion volumes exceed 20 units of PRBCs [23]. Previous studies have reported a massive transfusion prevalence of 12.3% during LT, which is associated with poor outcomes [24].

Our findings align with others that report a strong inverse correlation between platelet transfusions and outcomes [15,25–28]. Although FFP transfusions aim to correct coagulation defects, they often fail to reduce the need for PRBC transfusions because of their limited efficacy in restoring thrombin generation [29] and may increase bleeding through volume expansion [28,30,31]. Despite these limitations, FFP remains essential for correcting coagulation defects in end-stage liver disease, while platelets play critical role in managing thrombocytopenia and surgical bleeding during LT [26,32]. However, the optimal transfusion ratio of PRBCs, FFP, and platelets in LT remains unknown [21]. Evidence from trauma studies suggest that lower PRBC: FFP and PRBC: platelet ratios may yield better outcomes [33–36]. Specifically, in patients receiving UMT, maintaining an PRBC: FFP: platelet transfusion ratio below 1.5: 1: 1 has been linked to improved survival [4]. These practices have been well-established in trauma settings; however, the distinct pathophysiology of chronic liver disease and the complexity of LT techniques indicate that the optimal transfusion ratios in trauma and LT are different.

In LT, restrictive transfusion strategies have been advocated to enhance patient outcomes, reduce postoperative complications, and improve survival [9,15,23]. Recent studies have focused on applying patient blood management principles in LT, promoting the judicious use of blood products through real-time coagulation monitoring tools such as TEG and rotational thromboelastometry, along with protocols for managing PRBC-to-plasma ratios and fluid restriction [9,14,17,19,28,37]. Using this strategy, we observed the novel 4: 2: 1 PRBC: FFP: platelets transfusion ratio in our cohort, which departs from conventional transfusion practices in LT and aligns with recent efforts to tailor transfusion protocols to individual patient needs.

## Study implications

The management of LT patients requires addressing the complexity of preexisting coagulopathy, portal hypertension, and hypercoagulability risks, often necessitating complex transfusion strategies. The 4: 2: 1 PRBC: FFP: platelet transfusion ratio observed in this study reflects a pragmatic response to these challenges. Guided by point-of-care tools such as TEG, this approach enables individualized, real-time transfusion adjustments. Further, our findings imply that the 4: 2: 1 transfusion ratio was not associated with an increase in morbidity or mortality. This balance of risks and benefits suggests that this ratio may be a pragmatic transfusion strategy for LT, in the absence of point-of-care viscoelastic testing.

## Strengths and limitations

Our study has several strengths. It is the first to focus specifically on UMFT in LT, providing valuable insights into the outcomes associated with high-volume resuscitation. The detailed follow-up allowed for a thorough evaluation of postoperative recovery, complications, and transfusion practices and their links to patient outcomes. Additionally, the granular data collected on LT-specific complications such as graft non-function, hepatic artery thrombosis, and bile leaks, enhance the understanding of clinical challenges in this setting. Conducted at a high-volume, experienced transplant center, this study reflects advanced clinical practices within a specialized environment.

However, this study has certain limitations. The lack of propensity matching for non-UMFT cases limits direct comparisons. The absence of a non-UMFT group also precludes meaningful statistical comparison of predictor variables for UMFT. The results may have been influenced by local protocols and staff practices, which may have affected their broader applicability. Therefore, multicenter validation on the transfusion ratios used in this study to optimize UMFT implementation is warranted. These findings are specific to LT and may not be generalizable to other surgeries, pediatric patients, or centers lacking point-of-care TEG. Future multicenter studies with matched methodologies and broader patient populations are also required to validate these findings.

The estimated effects of the PRBC to FFP, platelet and cryoprecipitate ratios on many of the complications may be confounded by factors such as multicollinearity and the limited sample size relative to the incidence of these specific outcomes. This is evidenced by the extremely high (infinite) values of the estimates for some of the outcome variables as seen in the supplementary tables. For instance, cryoprecipitate transfusion is also a consequence of severe intra- and postoperative bleeding, which is independently associated with prolonged mechanical ventilation and extended hospital stay. Therefore, these specific findings should be interpreted with caution, and we recommend focusing on the significant trends rather than the absolute magnitude of the estimated effects.

The statistically significant findings regarding the relationship between intraoperative and postoperative fluid volumes and postoperative outcomes warrant cautious interpretation. While elevated fluid or transfusion volumes may contribute to adverse outcomes, reverse causality cannot be definitively excluded from consideration. Patients experiencing prolonged ICU or hospital stays due to complications unrelated to hemorrhage (such as sepsis or organ dysfunction) may have received additional fluids or blood products as part of their supportive care regimen. This bidirectional relationship inherently complicates causal inference in our statistical regression models. Although we implemented a sensitivity analysis, this approach also presents methodological limitations. The "resuscitation shift" parameter, as shown in S4 and S5 Tables indicates that fluid administration or blood transfusion at different perioperative phases represents distinct physiological responses that reflect variations in bleeding patterns, hemodynamic stability, and resuscitation requirements. However, identical "resuscitation shift" values (e.g., resuscitation shift = 0) may represent fundamentally different clinical scenarios-for instance, minimal intraoperative plus minimal postoperative volumes versus substantial intraoperative plus substantial postoperative volumes. The former scenario would reflect outcomes associated with conservative resuscitation strategies, whereas the latter would demonstrate outcomes related to aggressive volume replacement approaches. A suboptimal model fit across multiple regression analyses therefore warrants cautious interpretation. Potential contributors

to our model instability include limited sample size, imbalance between complicated and uncomplicated cases, and the absence of a control group for comparative assessment. Consequently, our granular analyses of the relationship between intraoperative and postoperative fluid volumes and outcomes are susceptible to bias. Therefore, inferences should prioritize the magnitude of associations between fluid/transfusion volumes and outcomes rather than precise effect estimates.

## Conclusions

UMFT is required in a substantial proportion of patients undergoing LT, reflecting not only the high demand for perioperative blood products and fluid resuscitation but also the complexity of these patients. Although UMFT was associated with prolonged recovery, it was not associated with increased complications, graft failure, or mortality. These findings suggest that with precautions and tailored, well-structured resuscitation to balance bleeding and clotting, safe recovery can be achieved without compromising patient outcomes while ensuring efficient resource utilization in LT. The observed 4: 2: 1 PRBC: FFP: platelet transfusion ratio represents a pragmatic approach to managing these challenges and could serve as a basis for optimizing transfusion practices in LT. Future research should aim to validate these findings through multicenter studies, develop predictive risk models for UMFT, and refine transfusion protocols to further improve outcomes in this high-risk population.

## Supporting information

**S1 Table. Missing preoperative and intraoperative laboratory results across stages of liver transplant operation.**
(DOCX)

**S2 Table. Preoperative and intraoperative laboratory results.**
(DOCX)

**S3 Table. Postoperative complications by Clavien-Dindo classification of surgical complications in liver transplantation patients.**
(DOCX)

**S4 Table. Associations between intraoperative and postoperative fluid/transfusion volumes and postoperative recovery outcomes, including resuscitation shift.**
(DOCX)

**S5 Table. Associations between intraoperative and postoperative fluid/transfusion volumes and postoperative complications, including resuscitation shift.**
(DOCX)

**S6 Table. Adjusted relationship between intraoperative and first 24-hour postoperative PRBCs transfusion and postoperative complications.**
(DOCX)

**S7 Table. Adjusted relationship between intraoperative and first 24-hour postoperative resuscitation volume and postoperative complications.**
(DOCX)

**S8 Table. Impact of FFP:PRBC and Platelet:PRBC ratios on recovery outcomes in liver transplantation patients.**
(DOCX)

**S9 Table. Impact of FFP: PRBC ratios on complications in liver transplantation patients.**
(DOCX)

**S10 Table. Impact of Platelet: PRBC ratios on complications in liver transplantation patients.**
(DOCX)

**S11 Table. Impact of Cryoprecipitate: PRBC ratios on complications in liver transplantation patients.**
(DOCX)

**S1 File. De-identified database.**
(DOCX)

## Acknowledgments

None.

## Author contributions

**Conceptualization:** Nattaya Raykateeraroj, Dong-Kyu Lee, Laurence Weinberg.

**Data curation:** Hugh Slifirski, Angelica Armellini, Riley Hazard, Jordan Zalcman, Junyan Zhao, Zac Tran, Peter Le, Wendell Zhang, Laurence Weinberg.

**Formal analysis:** Nattaya Raykateeraroj, Dong-Kyu Lee.

**Investigation:** Nattaya Raykateeraroj, Michael Fink, Marcos Vinicius Perini, Anoop N Koshy, Dong-Kyu Lee, Laurence Weinberg.

**Methodology:** Nattaya Raykateeraroj, Michael Fink, Marcos Vinicius Perini, Anoop N Koshy, Dong-Kyu Lee, Laurence Weinberg.

**Project administration:** Laurence Weinberg.

**Resources:** Laurence Weinberg.

**Supervision:** Michael Fink, Marcos Vinicius Perini, Anoop N Koshy, Laurence Weinberg.

**Validation:** Nattaya Raykateeraroj, Dong-Kyu Lee, Laurence Weinberg.

**Visualization:** Nattaya Raykateeraroj, Dong-Kyu Lee, Laurence Weinberg.

**Writing – original draft:** Hugh Slifirski, Nattaya Raykateeraroj, Angelica Armellini, Riley Hazard, Jordan Zalcman, Junyan Zhao, Zac Tran, Peter Le, Wendell Zhang, Dong-Kyu Lee, Laurence Weinberg.

**Writing – review & editing:** Hugh Slifirski, Nattaya Raykateeraroj, Angelica Armellini, Riley Hazard, Jordan Zalcman, Junyan Zhao, Zac Tran, Peter Le, Wendell Zhang, Michael Fink, Marcos Vinicius Perini, Anoop N Koshy, Dong-Kyu Lee, Laurence Weinberg.

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
