## [Decision Letter · Decision Letter 0]

May 17 2025

Dear Dr. Weinberg,

Thank you for submitting your manuscript to PLOS ONE. After careful consideration, we feel that it has merit but does not fully meet PLOS ONE’s publication criteria as it currently stands. Therefore, we invite you to submit a revised version of the manuscript that addresses the points raised during the review process.

We look forward to receiving your revised manuscript.

Kind regards,

Ryan M. Thomas, MD

Academic Editor

PLOS ONE

Journal Requirements:

Reviewers' comments:

Reviewer's Responses to Questions

**Comments to the Author**

1. Is the manuscript technically sound, and do the data support the conclusions?

Reviewer #1: Yes

2. Has the statistical analysis been performed appropriately and rigorously?

Reviewer #1: Yes

3. Have the authors made all data underlying the findings in their manuscript fully available?

Reviewer #1: Yes

4. Is the manuscript presented in an intelligible fashion and written in standard English?

Reviewer #1: Yes

Reviewer #1: The authors present a single-center retrospective observational study aimed at determining the prevalence of ultra-massive fluid transfusion (defined as >20 liters of crystalloids, colloids, blood, and blood products administered intraoperatively and within the first 24 hours postoperatively) in patients undergoing liver transplantation and examining its association with clinical outcomes. This study provides useful information on the prevalence of ultra-massive fluid transfusion in liver transplantation and its association with prolonged recovery. However, to improve the rigor and utility of the findings, the authors should address a number issues:

A significant limitation of the study is the lack of analysis on predictors of massive transfusion. Given the challenges in managing patients requiring ultra-massive transfusions, it would be beneficial to identify predictors of massive transfusion and the volume of transfusion required. This could include previously identified predictors such as those reported by Rana et al.(Ref 23), which include previous abdominal operations, warm ischemia time, hepatectomy time, and bilirubin levels.

The study does not provide details on the timing of fluid administration during the perioperative period. Differentiating when patients received fluids—such as up to hepatectomy, post-reperfusion, and postoperatively—would offer insights into specific phases of surgery where fluid management is most critical. Detailed reporting on this aspect would enhance understanding of fluid administration's impact, as well as its causes, at various surgical stages. A single center study with access to granular data should have this as its major advantage over a larger multicenter or national database study

While the study evaluated the relationship between FFP to PRBC ratio and platelet to PRBC ratio, it did not investigate the cryoprecipitate to PRBC ratio. Given the importance of cryoprecipitate in managing coagulopathy, this omission limits the comprehensiveness of the study's analysis.

The use of antifibrinolytic agents such as tranexamic acid or amicar was not explored. These medications are crucial in managing bleeding and could significantly impact transfusion requirements and clinical outcomes.

The study did not address the potential cardiac effects of ultra-massive transfusion, especially right heart failure.

The precision of the reported data in Table 2 is excessive. For example, a total fluid volume reported as 31917.7 mL implies a level of precision that is not practical given the inherent variability and error in clinical measurements. Similarly, the total norepinephrine use in Table 3 should be presented with appropriate rounding to reflect real-world clinical practice more realistically.

**Do you want your identity to be public for this peer review?** For information about this choice, including consent withdrawal, please see our Privacy Policy

Reviewer #1: No

---

## [Author Response · Author response to Decision Letter 1]

20 Apr 2025

My coauthors and I would like to thank you and the expert reviewers for taking the time to provide a very constructive and thoughtful review of our manuscript. We thank the reviewer for their expert insights and time spent reviewing our manuscript. We have provided a detailed comment to each of the reviewer’s questions, as outlined below.

1. Reviewer 1, comment 1: A significant limitation of the study is the lack of analysis on predictors of massive transfusion. Given the challenges in managing patients requiring ultra-massive transfusions, it would be beneficial to identify predictors of massive transfusion and the volume of transfusion required. This could include previously identified predictors such as those reported by Rana et al. (Ref 23), which include previous abdominal operations, warm ischemia time, hepatectomy time, and bilirubin levels.

Authors’ response to R1 comment 1: Thank you for your feedback on the importance of predictors of massive transfusion.

We acknowledge the importance of identifying predictors of massive transfusion and transfusion volume, given the complexity inherent in managing these patients. However, the design of our study differs substantially from that of Rana et al. In their retrospective analysis of 233 consecutive liver transplant recipients, Rana and colleagues included both patients who did and did not receive a massive transfusion. Through this approach, they identified four significant risk factors for intraoperative blood requirements when comparing patients who required transfusion to those who did not: warm ischemia time (OR 1.12, 95% CI 1.06–1.18), bilirubin (OR 1.04, 95% CI 1.02–1.08), previous surgery (OR 1.7, 95% CI 1.02–2.9), and hepatectomy time (OR 1.01, 95% CI 1.00–1.02). Specifically, warm ischemia time, bilirubin, prior surgery, and hepatectomy time were 12%, 4%, 70%, and 1% higher, respectively, in the transfused group compared to the non-transfused group.

In our study we are unable to explore predictors of ultra-massive fluid transfusion (UMFT). The development of a valid prediction model for UMFT in our study requires the inclusion of both patients who received UMFT and those who did not. The absence of a “control group” precludes meaningful statistical comparison of predictor variables, thereby impeding any model’s ability to distinguish factors truly associated with UMFT from random variation. Given that our study only included cases of UMFT, this “imbalance” increases the risk of overfitting and undermines the reliability of the model in this liver transplant population. This has been briefly updated in the limitations section.

We appreciate the author’s insightful comment and agree that this represents an excellent direction for future research, which we are now actively exploring.

2. Reviewer 1, comment 2: The study does not provide details on the timing of fluid administration during the perioperative period. Differentiating when patients received fluids—such as up to hepatectomy, post-reperfusion, and postoperatively—would offer insights into specific phases of surgery where fluid management is most critical. Detailed reporting on this aspect would enhance understanding of fluid administration's impact, as well as its causes, at various surgical stages. A single center study with access to granular data should have this as its major advantage over a larger multicenter or national database study

Authors’ response to R1 comment 2: Thank you for this insightful suggestion. We concur that the timing of fluid administration is both informative and clinically significant. In response, we have conducted a comprehensive review of each medical record to accurately determine the timing of all fluids administered. This information is now presented as an additional figure (Figure 2), which graphically depicts the volume and type of fluids and blood products administered at each stage of liver transplantation—including induction of anesthesia, liver dissection phase, anhepatic phase, reperfusion, and neohepatic phase—as well as during the first 12 hours and the 12–24 hour period postoperatively.

The data clearly demonstrate that most of the fluid administration occurs intraoperatively, particularly during the liver dissection phase representing a period of particularly high fluid and blood product usage, consistent with its known association with increased bleeding risk.

We believe that this figure provides a comprehensive and visually accessible representation of perioperative fluid and blood product utilization, effectively illustrating patterns of administration across the various phases of liver transplantation and the immediate postoperative period.

We have uploaded Fig 2 as a new file.

3. Reviewer 1, comment 3: While the study evaluated the relationship between FFP to PRBC ratio and platelet to PRBC ratio, it did not investigate the cryoprecipitate to PRBC ratio. Given the importance of cryoprecipitate in managing coagulopathy, this omission limits the comprehensiveness of the study's analysis.

Authors’ response to R1 comment 3: Thank you for this insightful comment. We agree that investigating the cryoprecipitate (CPP) to packed red blood cell (pRBC) ratio is highly relevant in the context of massive transfusion during liver transplantation, given the critical role of cryoprecipitate in correcting coagulopathy.

In response, we have conducted a detailed analysis examining the relationship between the CPP:pRBC ratio and key clinical outcomes. As demonstrated by the linear regression results presented in the table below, a higher CPP ratio was significantly associated with a reduced duration of mechanical ventilation (coefficient = –99.42, 95% CI: –189.85 to –8.99, P = 0.039). Furthermore, the analysis revealed a significant interaction between CPP ratio and pRBC transfusion (coefficient = 10.89, 95% CI: 5.1 to 16.69, P = 0.001), indicating that the beneficial effect of a higher CPP ratio on reducing ventilation time diminishes as the volume of pRBC transfusion increases.

Regarding length of hospital stay, the CPP ratio alone was not significantly associated with length of hospital stay (P = 0.576). However, the interaction term between pRBC and CPP ratio showed a significant positive association (coefficient = 0.89, 95% CI: 0.04 to 1.73, P = 0.047), suggesting that patients receiving both higher volumes of pRBC and higher CPP ratios experienced longer hospital stays. No significant associations were observed between the CPP:pRBC ratio and ICU length of stay.

These findings have been incorporated into the Results section and are detailed in Table 8 and an additional Supplementary File (Supp File 9).

It is important to note, as highlighted in the table below, that the estimated effects of the pRBC:CPP ratio on outcomes such as “return to theatre for bleeding” may be confounded by factors such as multicollinearity and the limited sample size relative to the incidence of these specific outcomes. This is evidenced by the extremely high (infinite) values of the estimates for “return to theatre for bleeding.” For instance, CPP transfusion may be a consequence of severe intra- and postoperative bleeding, which could itself be associated with prolonged mechanical ventilation and extended hospital stay. Therefore, these specific findings should be interpreted with caution, and we recommend focusing on the significant trends rather than the absolute magnitude of the estimated effects.

We have also addressed these limitations in the relevant section of the manuscript.

4. Reviewer 1, comment 4: The use of antifibrinolytic agents such as tranexamic acid or amicar was not explored. These medications are crucial in managing bleeding and could significantly impact transfusion requirements and clinical outcomes.

Authors’ response to R1 comment 4: Thank you for this important comment.

We have reviewed each medical record. In total 81 patients (1005) received tranexamic acid (2 grams) during the dissection or anhepatic phase. Five patients (61.7%) had evidence of hyperfibrinolysis on point-of-care TEG during the neohepatic phase and each received an additional 2 grams of tranexamic acid. This information has now been incorporated into the Results section.

We did not perform statistical analysis on the use of tranexamic acid, as all patients received at least one dose and only a small subset (n = 5) received an additional dose. The limited variability in tranexamic acid administration precludes meaningful statistical analysis regarding its impact on bleeding, fluid, or blood product requirements due to insufficient statistical power.

Reviewer 1, comment 5: The study did not address the potential cardiac effects of ultra-massive transfusion, especially right heart failure.

Authors’ response to R1 comment 5: Thank you for this very important and insightful comment.

At our centre, transesophageal echocardiography (TEE) is increasingly employed during adult liver transplantation (LT), particularly in the context of massive transfusion, due to its capacity to provide real-time, dynamic assessment of cardiac function, volume status, and intraoperative complications. Our practice is consistent with recommendations from multiple national and international anaesthesiology societies, as well as recent systematic reviews, which advocate for TEE monitoring during LT to facilitate dynamic cardiac assessment and guide clinical interventions.

Continuous intraoperative visualization of cardiac chambers and function via TEE enables the immediate identification of the etiology of hemodynamic instability, such as hypovolemia, right ventricular (RV) failure, cardiac tamponade, or embolic phenomena. In our experience, TEE offers critical guidance for the administration of fluids and blood products, optimizing volume status and mitigating the risks of both hypovolemia and fluid overload—the latter of which may precipitate pulmonary edema and RV failure. Moreover, TEE facilitates the rapid diagnosis of intraoperative complications, including pulmonary embolism, intracardiac thrombus, pericardial effusion, and air embolism, thereby enabling prompt and targeted intervention.

In our practice for liver transplantation, TEE is utilized for both qualitative and quantitative assessments of right and left ventricular function. Quantitative indices such as tricuspid annular plane systolic excursion (TAPSE) less than 1.6 cm and RV fractional area change (RVFAC) below 35% are employed to identify RV dysfunction. We favor TAPSE for its ease of measurement and reproducibility, while RVFAC correlates strongly with RV ejection fraction and is sensitive to changes in RV systolic performance. During episodes of massive transfusion, visual assessment of RV size (dilatation and hypokinesis indicating RV overload or failure), right atrial enlargement (suggestive of RV dysfunction), and abnormal septal motion (paradoxical or flattened motion indicative of RV pressure overload) provides immediate evidence of RV dysfunction or acute reperfusion events.

In our recent case series, intraoperative TEE was performed in 65 patients (80%). TEE was withheld in 16 patients (12%), primarily due to the presence of grade 3 or actively bleeding esophageal varices (n = 12), esophageal strictures (n = 2), esophageal diverticulum (n = 1), or recent esophageal surgery (n = 1). Intraoperative evidence of RV dysfunction was identified in six patients; of these, one patient died intraoperatively, while RV function normalized postoperatively in the remaining five patients.

These findings have been incorporated into the manuscript. We anticipate that the inclusion of these results will enhance understanding of the interplay between massive transfusion and perioperative cardiac dysfunction in liver transplantation.

References

1. De Marchi L, Wang CJ, Skubas NJ, et al. Safety and Benefit of Transesophageal Echocardiography in Liver Transplant Surgery: A Position Paper From the Society for the Advancement of Transplant Anesthesia (SATA). Liver Transpl. 2020 Aug;26(8):1019-1029. doi: 10.1002/lt.25800. PMID: 32427417.

2. Hansebout C, Desai TV, Dhir A. Utility of Transesophageal Echocardiography During Orthotopic Liver Transplantation: A Narrative Review. Annals of Cardiac Anaesthesia 26(4):p 367-379, Oct–Dec 2023. | DOI: 10.4103/aca.aca_186_2

3. Gouvêa G, Feiner J, Joshi S, et al. (2022) Evaluation of right ventricular function during liver transplantation with transesophageal echocardiography. PLoS ONE 17(10): e0275301. https://doi.org/10.1371/journal.pone.0275301 [8][10].

4. De Pietri L, Mocchegiani F, Leuzzi C, et al. Transoesophageal echocardiography during liver transplantation. World J Hepatol 2015; 7(23):24322448 [PMID:26483865DOI:10.4254/wjh.v7.i23.2432]

5. Reviewer 1, comment 6: The precision of the reported data in Table 2 is excessive. For example, a total fluid volume reported as 31917.7 mL implies a level of precision that is not practical given the inherent variability and error in clinical measurements.

Authors’ response to R1 comment 6: Thank you for this excellent suggestion. We agree that the precision reported in some of the tables is excessive and implies a level of precision that is not practical and is prone to error in clinical measurement. These have all been converted to Litres. This has been corrected throughout the manuscript, inclusive of all Tables.

6. Reviewer 1, comment 7: Similarly, the total norepinephrine use in Table 3 should be presented with appropriate rounding to reflect real-world clinical practice more realistically.

Authors’ response to R1 comment 7: Thank you again for this suggestion, as above, we have adjusted Table 3 accordingly.

Once again, our most sincere thanks to the Editor for dedicating their valuable time and expertise to review our manuscript. We deeply appreciate your thoughtful feedback and constructive suggestions, which have greatly contributed to improving the clarity and quality of our work. Your guidance has been instrumental in helping us refine our review, and we are grateful for the opportunity to benefit from your insights.

We hope that our manuscript is now suitable for publication in PLoS One. We would welcome the opportunity to make further modifications if needed.

Sincerely,

Professor Laurence Weinberg

BSc, MBBCh, MRCP, DipCritCareEcho, FANZCA, MD, PhD, ExecDipOL(Oxford)

Director, Department of Anaesthesia, Austin Hospital

Professor, Department of Critical Care, University of Melbourne, Australia

---

## [Decision Letter · Decision Letter 1]

Jun 13 2025

Dear Dr. Weinberg,

Thank you for submitting your manuscript to PLOS ONE. After careful consideration, we feel that it has merit but does not fully meet PLOS ONE’s publication criteria as it currently stands. Therefore, we invite you to submit a revised version of the manuscript that addresses the points raised during the review process.

We look forward to receiving your revised manuscript.

Kind regards,

Ryan M. Thomas, MD

Academic Editor

PLOS ONE

Journal Requirements:

Reviewers' comments:

Reviewer's Responses to Questions

**Comments to the Author**

Reviewer #1: (No Response)

2. Is the manuscript technically sound, and do the data support the conclusions?

Reviewer #1: Yes

3. Has the statistical analysis been performed appropriately and rigorously?

Reviewer #1: Yes

4. Have the authors made all data underlying the findings in their manuscript fully available?

Reviewer #1: (No Response)

5. Is the manuscript presented in an intelligible fashion and written in standard English?

Reviewer #1: Yes

Reviewer #1: The authors have performed much of the additional data collection and analyses requested. I also accept their rationale for not performing an analysis of the predictors of UMFT.

There are two subanalyses from the newly revealed data that would be beneficial to the manuscript.

1. The authors have done the work to parse the time period during which the fluids were given to the patients. Was there any specific period of time when volume was administered that was most correlated with outcomes? Or was there only a significant correlation when looking at total volume? It would be important to know whether intraop versus postop fluid was most correlated, or perhaps even prehepatectomy versus post reperfusion.

2. Similarly, given the number of episodes of right heart failure they observed, were there correlations with intraoperative fluid administration? While there may be too few events to do proper statistics, some further detail would be helpful to understand the scenarios.

**Do you want your identity to be public for this peer review?** For information about this choice, including consent withdrawal, please see our Privacy Policy

Reviewer #1: No

---

## [Author Response · Author response to Decision Letter 2]

15 May 2025

PLEASE REFER to uploaded "Response to Reviewers" Document that outlines all the formatted Tables.

Ryan M. Thomas, M.D

Academic EditorPLoS ONE

Dear Dr. Thomas

Ref: Resubmission of PONE-D-25-06585R1

Title: Ultra-massive fluid transfusion in adult liver transplant recipients: a single center observational study

Once again, my coauthors and I would like to thank you and the expert reviewers for taking the time to provide a very constructive and thoughtful review of our manuscript.

We deeply appreciate your thoughtful feedback and constructive suggestions, which have contributed to improving the clarity and quality of our work. We are pleased that there are now minor revisions that have been requested. Accordingly, we have provided a detailed response to the important points raised by Reviewer 1.

As requested, we have included the following items with our resubmission:

1. A point-by-point response letter (to follow) that addresses each issue raised by the Academic Editor and the Reviewers. I have uploaded this letter as a separate file labelled “Response to Reviewers.”

2. A marked-up copy of the manuscript that highlights changes made to the original version. This has been uploaded as a separate file labelled “Revised Manuscript with Tracked Changes.”

3. An unmarked version of our revised paper without tracked changes. This has also been uploaded as a separate file labelled “Manuscript.”

Comments by Reviewer 1

1. Reviewer 1, comment: The authors have performed much of the additional data collection and analyses requested. I also accept their rationale for not performing an analysis of the predictors of UMFT. There are two subanalyses from the newly revealed data that would beneficial to the manuscript.

We sincerely appreciate your expert insights, and the time dedicated to reviewing our manuscript. As a result of your thoughtful feedback, the manuscript has been significantly strengthened and improved. We are grateful for the opportunity to address the two questions raised below.

Reviewer 1, Question 1: The authors have done the work to parse the time period during which the fluids were given to the patients. Was there any specific period of time when volume was administered that was most correlated with outcomes? Or was there only a significant correlation when looking at total volume? It would be important to know whether intraop versus postop fluid was most correlated, or perhaps even prehepatectomy versus post reperfusion.

Authors’ response to Question 1: Thank you for this every insightful question. We have now undertaken further analysis investigating the associations between intraoperative and postoperative fluid/transfusion volumes and postoperative recovery outcomes. In addition, we have further the evaluated the associations between intraoperative and postoperative fluid/transfusion volumes and postoperative complications. These additional analyses have been included in the revised manuscript.

As seen in Table 1 below, we have used the term “resuscitation shift”, which we have operationally defined as the net change in postoperative fluid or transfusion volume administered compared to the volume of fluid administration intraoperatively. Two regression models were adjusted for perioperative covariates to analyze these relationships. Models for intraoperative and postoperative variables of fluid or transfusion were fitted simultaneously to assess their independent associations with clinical outcomes. As a supplementary sensitivity analysis, the resuscitation shift model (which we have termed the “trajectory model”) quantified the net volume difference between the intraoperative and postoperative 24-hour periods.

Our findings show that intraoperative and postoperative administration of fluids, packed red blood cell (RBC) units, and other blood products demonstrated statistically significant associations with adverse postoperative recovery metrics, including duration of mechanical ventilation and length of stay in both intensive care units (ICU) and hospital settings (See Table 1 below). The magnitude of association was more pronounced for postoperative fluid volumes, packed RBC units, and transfusions compared to their intraoperative counterparts, as evidenced by larger regression coefficients. The negative coefficients observed for resuscitation shift parameters suggest that increased postoperative fluid administration and transfusion (represented by smaller shift values) correlated with poorer clinical outcomes, thus highlighting the potential deleterious effects of excessive fluid or blood product administration during the early postoperative period.

These statistically significant findings warrant cautious interpretation. While elevated fluid or transfusion volumes may contribute to adverse outcomes, reverse causality cannot be definitively excluded from consideration. Patients experiencing prolonged ICU or hospital stays due to complications unrelated to hemorrhage (such as sepsis or organ dysfunction) may have received additional fluids or blood products as part of their supportive care regimen. This bidirectional relationship inherently complicates causal inference in statistical regression models. Although we implemented trajectory modeling as a sensitivity analysis, this approach also presents methodological limitations. The resuscitation shift parameter indicates that fluid administration or transfusion at different perioperative phases represents distinct physiological responses that reflect variations in bleeding patterns, hemodynamic stability, and resuscitation requirements. However, identical resuscitation shift values (e.g., resuscitation shift = 0) may represent fundamentally different clinical scenarios-for instance, minimal intraoperative plus minimal postoperative volumes versus substantial intraoperative plus substantial postoperative volumes. The former scenario would reflect outcomes associated with conservative resuscitation strategies, whereas the latter would demonstrate outcomes related to aggressive volume replacement approaches.

Table 2 below shows our analysis of discrete fluid volumes and transfusion quantities administered during intraoperative or postoperative 24-hour periods revealed that packed red blood cell (RBC) unit counts demonstrated a statistically significant association with both surgery-specific complications and severe postoperative complications (Clavien-Dindo Classification [CDC] grade ≥ 3). Postoperative RBC transfusion was associated with a lower risk of surgery-specific complications (odds ratio [OR] = 0.35, 95% confidence interval [CI]: 0.13–0.97, p = 0.043), potentially reflecting therapeutic benefits from timely correction of postoperative anemia or active hemorrhage. However, only 15.8% of surgery-specific complications in this cohort were attributable to bleeding, necessitating further granular data to clarify the role of postoperative transfusion in mitigating such outcomes.

Table 1. Associations between intraoperative and postoperative fluid/transfusion volumes and postoperative recovery outcomes, including resuscitation shift. (PLEASE SEE LETTER TO REVIEWERS)

The term resuscitation shift was operationally defined as the net change in postoperative fluid or transfusion volume administered compared to the volume of fluid administration intraoperatively. Two regression models were adjusted for perioperative covariates to analyze these relationships. Models for intraoperative and postoperative variables of fluid or transfusion were fitted simultaneously to assess their independent associations with clinical outcomes. As a supplementary sensitivity analysis, the resuscitation shift model (termed the trajectory model) quantified the net volume difference between the intraoperative and postoperative 24-hour periods. *: P < 0.05.

Table 2. Associations between intraoperative and postoperative fluid/transfusion volumes and postoperative complications, including resuscitation shift. (PLEASE SEE LETTER TO REVIEWERS)

The term resuscitation shift was operationally defined as the net change in postoperative fluid or transfusion volume administered compared to the volume of fluid administration intraoperatively. Two logistic regression models were adjusted for perioperative covariates to analyze these relationships. Models for intraoperative and postoperative variables of fluid or transfusion were fitted simultaneously to assess their independent effects. As a supplementary sensitivity analysis, the resuscitation shift model (termed the trajectory model) quantified the net volume difference between intraoperative and postoperative 24-hour periods. –: denotes model convergence failure or insufficient data for reliable estimation. These models were excluded from interpretation due to instability or low event rates. *: P<0.05.

While we thank the reviewer for this important comment, suboptimal model fit across multiple regression analyses warrants cautious interpretation. Potential contributors to model instability include limited sample size, imbalance between complicated and uncomplicated cases, and the absence of a control group for comparative assessment. Consequently, our granular analyses may be susceptible to bias, and inferences should prioritize the magnitude of associations between fluid/transfusion volumes and outcomes rather than precise effect estimates.

To enhance methodological transparency, we have addressed these limitations explicitly and would prefer relegating our detailed sensitivity analyses to supplementary materials contingent upon editorial request. We hope that our response to this question maintains analytical rigor while aligning with standards of academic discourse.

Reviewer 1, Question 2: Similarly, given the number of episodes of right heart failure they observed, were there correlations with intraoperative fluid administration? While there may be too few events to do proper statistics, some further detail would be helpful to understand the scenarios.

Authors’ response to Question 2: Thank you for this question. We acknowledge the Reviewer’s suggestion regarding the association between right cardiac dysfunction and intraoperative fluid administration. However, we must respectfully discuss that such an analysis is not statistically feasible with our current dataset. Specifically, intraoperative evidence of right ventricular dysfunction was identified in only six patients within our entire cohort, representing too few events to conduct meaningful statistical analysis or draw valid conclusions. Despite this limitation, we have performed a comparative assessment of these six patients with documented RV dysfunction against the remainder of the cohort, examining perioperative fluid volumes, use of blood products, and postoperative complications. As anticipated, this comparison revealed no statistically significant differences in any of these parameters between the groups. The paucity of events (n=6) substantially limits statistical power and increases the risk of both Type I and Type II errors, rendering any formal statistical inference unreliable.

While we appreciate the clinical relevance of this question, we believe that a properly powered study specifically designed to investigate this relationship would be required to generate meaningful conclusions about the association between right ventricular dysfunction and intraoperative fluid management strategies. We have noted this important limitation and potential avenue for future research in our discussion section.

Table 1: Comparison of fluid volumes and key outcomes for patients with intraoperative transesophageal features of intraoperative right cardiac dysfunction compared to patients without right sided cardiac dysfunction. (PLEASE SEE LETTER TO REVIEWERS)

Once again, our most sincere thanks to the Editor and Reviewer 1 for dedicating their valuable time and expertise to review our manuscript. We deeply appreciate your thoughtful feedback and constructive suggestions, which have greatly contributed to improving the clarity and quality of our work. Your guidance has been instrumental in helping us refine our review, and we are grateful for the opportunity to benefit from your insights.

We hope that our manuscript is now suitable for publication in PLoS One. We would welcome the opportunity to make further modifications if needed.

Sincerely,

Professor Laurence Weinberg

BSc, MBBCh, MRCP, DipCritCareEcho, FANZCA, MD, PhD, ExecDipOL(Oxford)

Director, Department of Anaesthesia, Austin Hospital

Professor, Department of Critical Care, University of Melbourne, Australia

---

## [Editor Report · Decision Letter 2]

Ultra-massive fluid transfusion in adult liver transplant recipients: a single center observational study

PONE-D-25-06585R2

Dear Dr. Weinberg,

We’re pleased to inform you that your manuscript has been judged scientifically suitable for publication and will be formally accepted for publication once it meets all outstanding technical requirements.

Kind regards,

Ryan M. Thomas, MD

Academic Editor

PLOS ONE